# Zinc and Diabetes: A Connection between Micronutrient and Metabolism

**DOI:** 10.3390/cells13161359

**Published:** 2024-08-15

**Authors:** Rahnuma Ahmad, Ronald Shaju, Azeddine Atfi, Mohammed S. Razzaque

**Affiliations:** 1Department of Physiology, Medical College for Women and Hospital, Dhaka 1230, Bangladesh; 2Department of Medical Education, School of Medicine, University of Texas Rio Grande Valley (UTRGV), Edinburg, TX 78541, USA; 3Department of Biochemistry and Molecular Biology, Massey Cancer Center, Virginia Commonwealth University, Richmond, VA 23298, USA

**Keywords:** diabetes mellitus, insulin sensitivity, zinc deficiency, zinc supplementation, glycemic control

## Abstract

Diabetes mellitus is a global health problem and a major contributor to mortality and morbidity. The management of this condition typically involves using oral antidiabetic medication, insulin, and appropriate dietary modifications, with a focus on macronutrient intake. However, several human studies have indicated that a deficiency in micronutrients, such as zinc, can be associated with insulin resistance as well as greater glucose intolerance. Zinc serves as a chemical messenger, acts as a cofactor to increase enzyme activity, and is involved in insulin formation, release, and storage. These diverse functions make zinc an important trace element for the regulation of blood glucose levels. Adequate zinc levels have also been shown to reduce the risk of developing diabetic complications. This review article explains the role of zinc in glucose metabolism and the effects of its inadequacy on the development, progression, and complications of diabetes mellitus. Furthermore, it describes the impact of zinc supplementation on preventing diabetes mellitus. The available information suggests that zinc has beneficial effects on the management of diabetic patients. Although additional large-scale randomized clinical trials are needed to establish zinc’s clinical utility further, efforts should be made to increase awareness of its potential benefits on human health and disease.

## 1. Introduction

Diabetes mellitus, which encompasses both type 1 and type 2 variants, is a metabolic disease characterized by diverse clinical presentations and disease progression patterns. Type 1 diabetes can be distinguished from type 2 diabetes by several characteristic features: (1) type 1 diabetes typically manifests at an age less than 35 years; (2) individuals with type 1 diabetes often have a BMI less than 25 kg/m^2^; (3) unintentional weight loss is commonly observed in those with type 1 diabetes, without any prior attempt to reduce weight; (4) at the time of presentation, individuals with type 1 diabetes may exhibit significantly elevated blood glucose levels, exceeding 20 mmol/L (360 mg/dL); and (5) the development of ketoacidosis, a life-threatening complication, is more common in individuals with type 1 diabetes [1]. Clinically, over 80% of diabetes cases are identified as type 2 diabetes mellitus. The dysfunction and reduction in the mass of pancreatic β-cells are critical factors in the initiation and progression of type 2 diabetes mellitus (Figure 1). This condition is also characterized by chronic hyperinsulinemia as a compensatory mechanism for insulin resistance.

According to the diagnostic criteria established by the American Diabetes Association, the following blood glucose parameters are considered for the diagnosis of diabetes mellitus: fasting plasma glucose ≥126 mg/dL (7.0 mmol/L) and no caloric intake for a minimum of 8 h. A plasma glucose level ≥200 mg/dL (11.1 mmol/L) was measured after a 75-g glucose load during an oral glucose tolerance test (OGTT). When HbA1c is ≥6.5% (48 mmol/mol), the HbA1c test must be performed using a method certified by the National Glycohemoglobin Standardization Program and standardized to the Diabetes Control and Complications Trial reference assay. A random plasma glucose level ≥200 mg/dL (11.1 mmol/L) was detected in individuals who presented with classic symptoms of hyperglycemia or hyperglycemic crisis [2]. Importantly, these diagnostic criteria are based on the measurement of plasma glucose or HbA1c, which serve as reliable indicators of the presence and severity of diabetes mellitus. The standardization of these laboratory tests, as specified by the guidelines, ensures the accuracy and consistency of the diagnostic process.

Alterations in the levels of trace elements, such as zinc, and an increase in oxidative stress, particularly in pancreatic β-cells, have been observed in individuals with diabetes mellitus. These imbalances can exacerbate insulin resistance in peripheral tissues such as muscle, adipose tissues, and liver, worsening glycemic control and developing diabetic complications. The progression of diabetes, in turn, can further deteriorate the overall status of zinc and other trace elements [3]. Zinc is an essential trace element that functions as a co-factor in the formation, storage, and release of insulin by the pancreas [4]. Reduced levels of zinc can impair the pancreas’s ability to synthesize and secrete insulin and diminish the uptake of glucose by peripheral cells, leading to the development of insulin resistance, a hallmark of diabetes mellitus [5]. The depletion of zinc observed in individuals with diabetes mellitus is primarily attributed to two key mechanisms: (1) impaired intestinal absorption of zinc (diabetes-associated gastrointestinal changes and alterations in zinc transport mechanisms can lead to reduced intestinal absorption of this essential micronutrient) and (2) increased urinary excretion of zinc (hyperglycemia and associated metabolic disturbances in diabetes can result in increased urinary loss of zinc, further exacerbating the deficiency) [4]. The interplay between trace element dysregulation and the pathogenesis of diabetes mellitus highlights the importance of maintaining optimal micronutrient status in the management of this chronic metabolic disorder. Addressing these underlying nutritional imbalances may offer a promising avenue for improving glycemic control and minimizing the risk of diabetic complications.

## 2. Objective of the Study and Methodology

This work aims to synthesize information from human and experimental studies to understand better the relationship between diabetes mellitus and the essential trace element zinc. Furthermore, this brief review describes the effects of zinc supplementation in the management of type 2 diabetes. The information was collected by searching the literature using databases such as Google Scholar, PubMed, and Scopus between November 2023 and February 2024. The keywords used to guide the search were ‘zinc in diabetes mellitus’, ‘zinc supplementation’, ‘zinc deficiency’, and ‘diabetic complications and zinc’.

By comprehensively analyzing the available evidence from both human and experimental studies, the critical roles of zinc in glucose metabolism and the pathogenesis of diabetes mellitus have been elucidated. The findings also highlight the potential benefits of zinc supplementation in managing type 2 diabetes and its associated complications. By applying this knowledge, we can work towards better clinical outcomes and quality of life for individuals with diabetes.

## 3. Sources of Zinc

Zinc is the second most abundant micronutrient found in nature after iron [6,7,8,9]. The dietary sources of zinc include both animal- and plant-based foods. Meat, seafood, whole grains, and oil seeds are the primary dietary sources of zinc [10]. The recommended dietary allowance (RDA) for zinc is 11 mg/day for adult males and 8 mg/day for adult females [11]. These amounts are considered adequate to support the activity of zinc-dependent metalloenzymes and maintain overall health. However, meeting the daily dietary need for zinc solely through food sources would require the consumption of excessive amounts of protein, dietary fiber, and fat, leading to a surplus of caloric intake and increased bowel movements due to the high fiber and fat content [10]. Some studies have reported slightly higher serum levels of zinc in women than men, although this difference is not always statistically significant [12].

The pharmacological dosage of zinc is considered to be more than 40 mg/day for individuals 19 years of age and older [11]. The chelated zinc doses used in various studies typically fall between 220 mg/day and 660 mg/day [13,14]. The forms of zinc commonly used for oral administration include zinc sulfate, zinc gluconate, zinc picolinate, and zinc citrate, as these forms are generally better absorbed than zinc oxide [15]. Consuming protein, such as whey protein, along with oral zinc supplementation, can also enhance the absorption of zinc [16].

## 4. Zinc Regulation

Zinc homeostasis is maintained through a combination of fractional absorption (approximately 20–40%) and excretion via the urine (0.5 mg/day) and intestine (1–3 mg/day) [17]. The range of zinc consumption that helps maintain a balance in the body’s zinc status is typically between 14 and 30 mg/kg [18,19]. However, zinc homeostasis can be maintained even with zinc intake as low as 2.8 mg/kg or as high as 40 mg/kg [18,19]. In response to such extreme levels of zinc intake, the body’s homeostatic mechanisms adjust the absorption and excretion of zinc to maintain adequate zinc status [20,21].

Zinc absorption primarily occurs in the small intestine [18]. The efficiency of zinc absorption from the diet depends on the zinc content and the composition of the diet, with aqueous zinc solutions being more efficiently absorbed in the fasted state (approximately 60–70%) than in the diet [18,22]. The average zinc absorption rate in humans is approximately 33% [18,23]. The absorption of zinc is also influenced by the concentration and status of zinc in the body [24]. During the digestive process, zinc is released as a free ion, which then binds to endogenously secreted ligands, including peptides and amino acids, metallothionein, histidine, and cysteine. These zinc—ligand complexes are transported into the duodenal and jejunal enterocytes [22,25]. Specialized transporters, such as zinc transporters (ZnTs) and ZIP transporters, facilitate the movement of zinc across the cell membrane and into the portal circulation. Zinc is then transported to the liver and released into the systemic circulation, primarily binding to plasma albumin (approximately 70%). Changes in serum albumin levels can, therefore, alter the circulating levels of zinc [22,25,26].

The distribution and expression of ZnTs are regulated by the body’s zinc status [27]. ZnTs that mediate the efflux of zinc into intracellular vesicles or out of the cell can decrease zinc availability within cells, whereas ZIP transporters that increase zinc uptake into cells and release zinc from intracellular vesicles can increase zinc availability (Figure 2) [28]. The expression and activity of these transporters are influenced by various factors, including cytokines (IL-1β, IFN-γ, and IL-6), hormones (insulin), zinc deficiency, and excess zinc [29]. With respect to zinc supplementation, the expression of ZnT1 in the intestinal villi has increased in animal studies [30,31]. It is currently understood that ZnT1 acts as a zinc exporter and regulates zinc homeostasis by facilitating the elimination or acquisition of zinc in response to excess or deficiency, respectively [31].

## 5. Zinc and Inflammation

Zinc is a crucial micronutrient that plays a vital role in the proper function of the immune system and the maintenance of overall health (Figure 3). It is essential for the proliferation and differentiation of immune cells [32] and potentially reduces viral infections [33,34]. Zinc has been found to bind nearly three thousand different proteins in the human body [35]. Many metalloenzymes require zinc for their regulatory and catalytic functions [35,36]. Zinc also serves as an important signaling molecule within the immune system and acts as a neuromodulator in synaptic vesicles. Several studies have indicated that zinc deficiency may contribute to the development of chronic and metabolic diseases, such as diabetes, neurodegenerative disorders, cancer, and intestinal diseases [6,37,38,39,40,41,42]. Adequate levels of zinc are essential for maintaining oral health [43] and for various hormonal functions [44]. Furthermore, zinc plays a role in lipid metabolism during obesity, often associated with type 2 diabetes. Studies have shown that zinc status can influence lipid profiles, the functionality of adipose tissue, adipokine production, and insulin sensitivity [45,46]. Earlier studies reported that zinc can exert insulin-like effects on adipocytes, influencing adipogenesis and glucose metabolism, and that zinc may play a role in enhancing insulin sensitivity in adipose tissues [47]. Zinc deficiency is a common clinical manifestation in individuals with obesity, perhaps thereby exacerbating metabolic dysregulation. Zinc supplementation has been shown to improve lipid profiles and potentially minimize some of the harmful consequences of obesity [48,49]. Obesity is frequently associated with lower serum levels of zinc in both genders than in their lean counterparts. Noteworthy, although childhood obesity is more common in boys than in girls, obesity becomes more common in adult women than in their male counterparts [50].

Zinc has been shown to aid in the regeneration of the intestinal epithelium, increase water and electrolyte absorption, increase the levels of brush border enzymes, and help the immune system clear pathogens through modulation [51,52,53]. Zinc exerts anti-inflammatory effects on human SW480 and HL-60 cell lines by inducing the expression of the A20 protein, an anti-inflammatory protein that suppresses the signaling pathways of Toll-like receptors (TLR) and tumor necrosis factor receptor (TNFR), thereby inhibiting the NF-κB pathway [54]. Studies conducted on ten normal healthy volunteers who consumed oral zinc supplementation (45 mg zinc gluconate) for eight weeks revealed that the expression of pro-inflammatory cytokines, such as TNF-α and IL-1β, was downregulated owing to the upregulation of the DNA-specific binding of A20 [55]. Zinc plays a role in hindering pro-inflammatory pathways. It prevents the nuclear translocation of NF-κB, modulates IκB kinase (which causes the phosphorylation of the NF-κB inhibitory protein), and inhibits the activation of signal transducer and activator of transcription 3 (STAT3) mediated by IL-6. Zinc also increases the level of cyclic guanosine monophosphate (cGMP) by inhibiting cyclic nucleotide phosphodiesterase (PDE). Studies conducted on human peripheral blood mononuclear cells (PBMCs) isolated from healthy donors have shown that an increase in cGMP activates protein kinase A (PKA), which in turn inactivates NF-κB and MAPK signaling [56]. Additionally, zinc can bind to and inhibit the translocation of protein kinase C to the cell membrane, indirectly inhibiting the activity of NF-κB, as shown in bone marrow-derived macrophages from mice [57].

Zinc also promotes anti-inflammatory pathways. In vitro studies have shown that zinc enhances the IL-2 signaling pathway by blocking MAP kinase phosphatases in the PTEN and ERK 1/2 pathways. This action counteracts the function of the PI3K/Akt pathway. Additionally, zinc aids in the movement and activity of Smad2 and Smad3 in the TGF-β signaling pathway, and it promotes the phosphorylation and movement of STAT6 into the nucleus via the IL-4 signaling pathway [38].

## 6. Zinc and Glucose Metabolism

Zinc plays a crucial role in the crystallization and signaling of insulin. Specifically, zinc promotes the activation of the PI3K/Akt pathway, which is essential for glucose metabolism [58]. As a cofactor, zinc has a critical function in the action of antioxidants and the metabolism of carbohydrates [45,59]. This micronutrient also aids in the phosphorylation of the β-subunit of the insulin receptor and the translocation of glucose transporter 4 (GLUT4) [60,61]. Importantly, insulin forms a hexameric structure by coupling with two zinc ions, a process necessary for the maturation of insulin within the secretory granules of pancreatic β-cells and the subsequent release of insulin [62,63] (Figure 4).

Pancreatic β-cells express specific zinc carrier proteins that play crucial roles in insulin secretion [64,65,66]. One such key protein is ZnT8, which is essential for the crystallization, processing, storage, and secretion of insulin, as well as the overall metabolism of glucose [67]. The ZnT8 transporter is responsible for shuttling zinc into the secretory granules of pancreatic β-cells, thereby facilitating the formation of the zinc-insulin hexamer, a critical step in insulin maturation and release [67,68,69,70]. Zinc deficiency leads to a reduction in the expression of ZnT8, ultimately impairing insulin secretion. In addition to ZnT8, other zinc transporter proteins, such as ZnT6 and ZnT5, are involved in the transport of zinc into the vesicles of pancreatic β-cells, where micronutrients participate in the metabolism of proinsulin and the subsequent secretion of mature insulin [62,67,71]. Furthermore, the zinc transporter protein ZnT7 is responsible for transporting zinc to the Golgi apparatus of pancreatic β-cells, an essential process for the proper formation of insulin [71,72].

Single nucleotide polymorphisms (SNPs) in the ZnT8 (SLC30A8) gene, particularly rs13266634, are associated with an increased risk of type 2 diabetes across various studied populations. The ZnT8 transporter is present in pancreatic β-cells and plays a crucial role in insulin production and release by relocating zinc into insulin-containing granules within these cells. The rs13266634 variant, along with other SNPs in this gene, can alter ZnT8 transporter functions to contribute to an increased risk of developing type 2 diabetes [73,74].

Zinc plays a crucial role in promoting glucose transport into cells. Specifically, zinc deactivates the enzyme tyrosine phosphatase 1B (PTP1B), which subsequently dephosphorylates the β-subunit of the insulin receptor, thereby preventing insulin signaling. Zinc also inhibits the enzyme tensin homolog (PTEN), which normally dephosphorylates phosphatidylinositol triphosphate and inhibits the Akt signaling pathway of insulin. By inhibiting PTEN, zinc induces the activity of Akt and PI3K, enhancing the responsiveness of insulin-regulated aminopeptidase (IRAP) and promoting the translocation of GLUT4 to the cell membrane in muscle and adipose tissue [61,71,75].

In addition to its role in glucose transport, zinc enhances glucose storage. Zinc stimulates the phosphorylation of glycogen synthase kinase 3 (GSK3) and the transcription factor forkhead box protein O1 (FoxO1). Phosphorylation of GSK3 promotes the activation of glycogen synthase, whereas phosphorylation of FoxO1 prevents it from stimulating the expression of gluconeogenic genes. Collectively, these actions of zinc help promote glycogen storage and inhibit glucose production [61,76]. Zinc promotes the uptake of glucose by cells, induces the expression of glucose transporter genes (GLUT1 and GLUT4), and regulates gluconeogenic enzymes such as glucose-6-phosphatase (G6Pase) and phosphoenolpyruvate carboxykinase (PEPCK), thereby improving glucose metabolism [5]. Consequently, zinc deficiency can lead to disturbances in glucose metabolism, potentially contributing to the development of diabetes mellitus. Conversely, hyperglycemia may interfere with the active transport of zinc into renal tubular cells, resulting in increased urinary zinc excretion [77]. Zinc deficiency in individuals with diabetes mellitus may also exacerbate chronic conditions, including diabetic polyneuropathy [78].

## 7. Zinc Deficiency and Diabetes: Experimental Studies

A significant increase in glucose levels was detected in streptozotocin-induced diabetic rats fed a zinc-deficient diet. These rats also presented with an increase in malondialdehyde, transaminase, triglyceride, and cholesterol. Additionally, the zinc levels in tissues such as the liver, pancreas, and femur decreased, whereas the activities of catalase, amylase, superoxide dismutase, glutathione-S-transferase, and lactate dehydrogenase decreased. The concentration of glutathione, a potent antioxidant, also decreased in diabetic rats fed a zinc-deficient diet [79].

In a separate study, diabetic rats fed a zinc-deficient diet for four weeks experienced further reductions in serum zinc levels, as well as decreases in testosterone and plasma insulin levels. The diabetic rats fed a zinc-deficient diet also presented increased triglyceride, cholesterol, and HbA1c levels. A zinc-deficient diet increased oxidative stress, as evidenced by a reduction in catalase, superoxide dismutase, and glutathione levels [80]. The study also reported changes in the levels of transcription factors (MTF-1, NF-κB) and enzymes (GPX4, GPX5), along with MT, a metal-binding protein, and Keap1, a regulatory protein that interacts with the transcription factor Nrf2, in the epididymides and testes of diabetic rats. Histopathological changes and abnormal alterations in sperm head morphology, sperm chromatin, protamine, and sperm decondensation were observed. The study concluded that zinc deficiency exacerbated the damage to germ cells in type 2 diabetes mellitus [80]. Another animal study by Sahu et al. investigated the role of zinc deficiency in the Bisphenol A (BPA)-induced toxicity of male germ cells and gonads in diabetic subjects. They reported that zinc deficiency in diabetic rats exacerbated BPA-induced toxicity in the epididymis and testes, as evidenced by structural damage, increased sperm abnormalities, DNA damage, and hypogonadism, compared with those in diabetic rats without zinc deficiency [81].

Researchers have reported widening Bowman’s space, thickening of the basement membrane, and loss of patches of apical microvilli in the proximal convoluted tubules in diabetic rats fed a standard diet. These changes were further aggravated, and degeneration was more extensive in diabetic rats fed a zinc-deficient diet, with a thickened cellular lining of both the proximal and distal tubules, swollen mitochondria, and the presence of secondary lysosomes (fusion of primary lysosomes with phagosomes). In contrast, the diabetic rats fed the zinc-supplemented diet presented minimal renal cortical changes. The expression of caspase-3, a mediator of cell apoptosis, was significantly higher in diabetic rats fed a standard diet and diabetic rats fed a zinc-deficient diet. Researchers have concluded that zinc deficiency exacerbates renal cortical structural alterations in diabetic rats [82]. Other similar studies have reported histological alterations due to zinc deficiency in diabetes, including interstitial fibrosis from excess collagen deposition, damage to endothelial cells, and the formation of hyaline masses [83,84,85,86,87,88,89]. Several other studies have shown that zinc deficiency can damage various organs, such as the retina, bone, liver, and pancreas, in the context of diabetes [83,86,90,91].

## 8. Zinc Deficiency and Diabetes: Clinical Studies

A cross-sectional study conducted in Riyadh, Saudi Arabia, between May 2014 and June 2015, involving 200 individuals with type 2 diabetes mellitus and 192 nondiabetic controls, revealed significantly lower mean serum zinc levels in diabetic individuals [66.54 ± 11.328 μg/dL) than nondiabetic individuals (82.63 ± 12.194 μg/dL, *p* < 0.001). Among the patients with diabetes mellitus, those with lower zinc levels had higher HbA1c levels (8.91 ± 2.16%) than those with normal zinc levels (5.696 ± 2.3, *p* < 0.001). The study considered normal zinc levels to be >70 μg/dL. A negative correlation (Pearson correlation coefficient ‘r’ = −0.527) was reported between serum zinc levels and fasting blood glucose and between serum zinc levels and HbA1c. The study concluded that low zinc levels were associated with type 2 diabetes mellitus and that zinc levels were negatively correlated with poor glycemic control [4].

Other studies, such as those conducted by Santa et al. [92], Sahria and Goswami [93], and Al-Maroof and Al-Sharbatti [94], have also reported significantly lower zinc levels in patients with diabetes than in healthy individuals (*p* < 0.001). The lower zinc levels observed in type 2 diabetic patients have been attributed to a decrease in zinc absorption from the gastrointestinal tract and an increase in urinary zinc excretion, as suggested by Marchesini et al. [95]. Several studies have reported a significant negative correlation between HbA1c and serum zinc levels in individuals with diabetes. Tripathy et al. reported a correlation ‘r’ = −0.408, indicating that higher HbA1c levels were associated with lower serum zinc concentrations. They also reported significantly lower zinc levels in diabetic individuals than nondiabetic individuals [77]. A similar negative association between zinc levels and HbA1c (‘r’ = −0.33) was noted by Al-Maroof and Al-Sharbatti [94]. Zinc deficiency may contribute to the development of diabetic complications, as zinc deficiency can exacerbate inflammation and ultimately lead to organ damage [78] (Figure 5). Human studies have reported the detrimental effects of zinc deficiency on various systems, including the nervous, renal, and reproductive systems. Hussein et al. conducted a study on 120 subjects, including 40 individuals with diabetic polyneuropathy, 40 diabetic subjects without polyneuropathy, and 40 healthy controls. They reported significantly lower zinc levels in individuals with and without neuropathy than in healthy subjects (*p* = 0.025 and *p* = 0.03, respectively) [78]. A significant negative association was also observed between serum zinc levels and the neuropathy symptom and change (NSC) score, Michigan neuropathy screening instrument (MNSI) score, and HbA1c level (*p* = 0.001, *p* = 0.003, and *p* < 0.001, respectively). Additionally, a significant association was found between serum zinc levels and nerve conduction [78]. Migdalis et al. suggested that zinc deficiency may lead to lipid peroxidation, which could result in the development of neuropathy in patients with diabetes [96]. 

## 9. Zinc Intervention and Diabetes: Clinical Outcome Studies

Zinc supplementation may aid in managing various clinical features of type 2 diabetes, including insulin resistance, hyperglycemia, and dyslipidemia [97]. Individuals with type 2 diabetes often exhibit hyperzincuria, hypozincemia, and impaired zinc absorption, leading to reduced zinc availability for the brain and increased production of inflammatory markers. This pro-inflammatory environment can also increase the susceptibility of these individuals to microbial infections [98,99].

Several animal and human studies have reported that zinc supplementation can improve fasting blood glucose levels, fasting insulin levels, and lipid profiles in type 1 and type 2 diabetes patients [95,100,101,102,103,104,105,106,107,108,109,110,111,112,113,114,115]. However, study results are not always consistent [116,117,118,119,120]. For instance, Foster et al. [116] found no significant improvements in lipid profiling with zinc supplementation in diabetic subjects, whereas a meta-analysis by Jayawardena et al. reported that zinc supplementation in individuals with diabetes led to improvements in glycemic control, blood pressure, and lipid profiles [114]. The studies utilized a wide range of zinc dosages, from 7.5 to 660 mg/day, with follow-up periods ranging from three weeks to six months. In a meta-analysis of 1700 participants from 32 interventional studies, zinc supplementation significantly reduced fasting glucose levels, with a weighted mean difference of −14.15 mg/dL [121]. Furthermore, supplementation reduced 2-h postprandial glucose levels by −36.85 mg/dL. The analysis also revealed decreases in fasting insulin and HbA1c levels, indicating improved glycemic control in individuals with diabetes [121]. The various zinc compounds used for supplementation include zinc chloride [118], zinc gluconate [119,120], zinc acetate [109,110], zinc phosphate, and zinc sulfate [95,101,102,103,104,105,106,107,108,111,112,113,115,122]. Zinc gluconate and zinc acetate are preferred for their good absorption and lower risk of digestive discomfort, making them suitable for general supplementation, whereas zinc sulfate is effective for treating deficiencies but may cause gastrointestinal issues. Zinc chloride and zinc phosphate have more specialized uses (dental applications and preservatives) and are less common in dietary supplementation. Choosing the right form of zinc compound depends on individual health needs, tolerance, and specific applications [15].

Notably, some studies have used zinc doses exceeding the recommended upper limit of 40 mg/day [106,107,110,112,113,115,119,120]. Such high-dose zinc supplementation may lead to copper deficiency and adversely affect the activity of antioxidant enzymes such as superoxide dismutase [123]. Additionally, zinc doses ≥150 mg/day may impair immune function [124,125]. Zinc supplementation can also cause iron deficiency in women with low iron stores [126]. A separate meta-analysis by Jafarnejad et al. suggested that a zinc dose of 20 mg/day may be optimal for improving metabolic parameters in individuals with diabetes [124]. The effects of zinc on diabetes mellitus are dose-dependent, with appropriate supplementation showing promise in improving glycemic control and insulin sensitivity. However, careful monitoring of zinc intake is necessary to avoid toxicity and maintain optimal metabolic health [127]. Despite our best efforts to present an unbiased summary of the role of zinc in diabetes, our reliance on available sources, which are not always derived from large cohort studies, could influence the objectivity of part of our conclusions. Most existing studies focus on the short-term effects of zinc supplementation. To fully understand its impact, it is also necessary to investigate the long-term effects. Further clinical and experimental studies are required to establish guidelines for optimal zinc supplementation in diabetes care.

## 10. Conclusions

Human and experimental studies have consistently demonstrated the beneficial role of zinc in the context of diabetes mellitus. Zinc plays a significant role in the proper functioning of β-cells, the action of insulin, and the homeostasis of glucose. Conversely, zinc deficiency can dysregulate glucose metabolism through the disruption of β-cell function, as well as the induction of oxidative stress and inflammation. Clinical studies have shown the positive impact of zinc supplementation on glycemic control in individuals with diabetes mellitus. Still, more large-scale randomized clinical trials with longer durations are needed to establish the safety and effectiveness of various forms of zinc supplementation in diabetic patients. Furthermore, efforts should be made to raise awareness among healthcare professionals regarding the benefits of a diet rich in micronutrients and adequate macronutrient intake. Promoting the consumption of a high-quality, micronutrient-dense diet may help reduce the development of inflammatory disorders among diabetic patients.

In summary, the available evidence strongly supports the critical role of zinc in glucose metabolism and the pathogenesis of diabetes mellitus. Addressing zinc and other essential mineral deficiencies through supplementation and dietary interventions may offer a promising avenue for the holistic management of this prevalent metabolic disorder and its complications [48,49,128,129]. 

## Figures and Tables

**Figure 1 cells-13-01359-f001:**
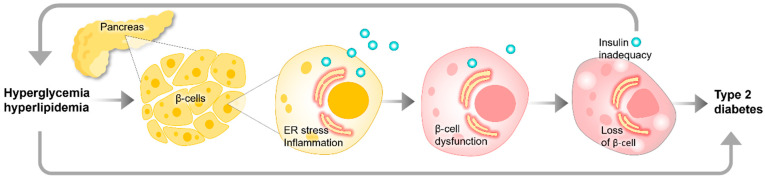
Simplified diagram of the evolution of type 2 diabetes, where metabolic disturbances lead to cellular stress, which damages insulin-producing β-cells. This damage, in turn, exacerbates the metabolic disturbances, creating a vicious cycle of disease progression.

**Figure 2 cells-13-01359-f002:**
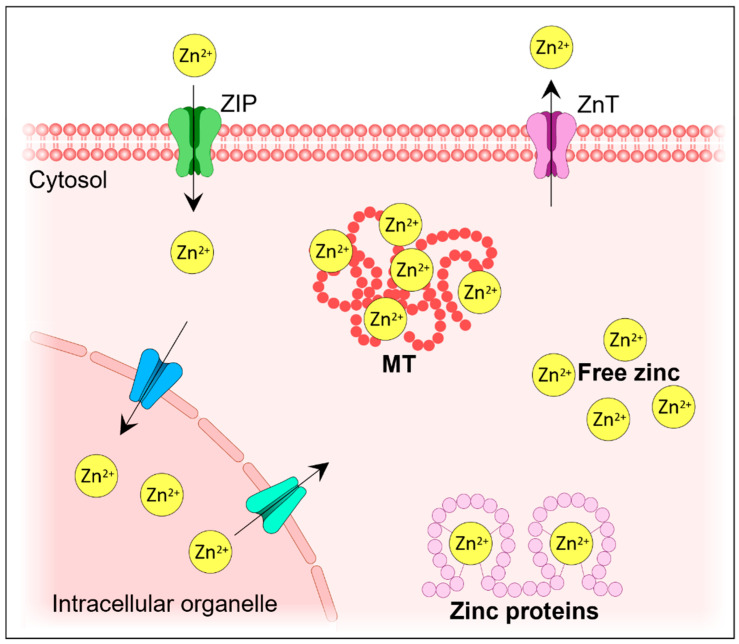
Cellular zinc homeostasis is maintained through the delicate interplay of three key protein groups: metallothioneins (MTs), Zrt-like and Irt-like proteins (ZIPs), and zinc transporters (ZnTs). The ZIP and ZnT families, which are specialized zinc transporter groups, coordinately regulate the movement of zinc ions in and out of the cytoplasm. Metallothioneins play crucial roles by binding them to zinc ions, serving as a zinc reserve, buffering zinc levels, and chelating excess zinc to prevent toxicity.

**Figure 3 cells-13-01359-f003:**
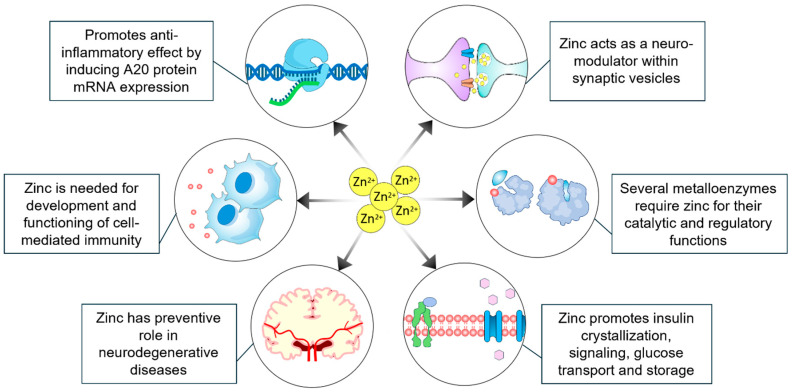
Zinc is an essential micronutrient that plays a critical role in human health by exerting positive effects on immune system function, maintaining cellular homeostasis, and delaying neurodegenerative and infectious diseases to maintain overall health and well-being.

**Figure 4 cells-13-01359-f004:**
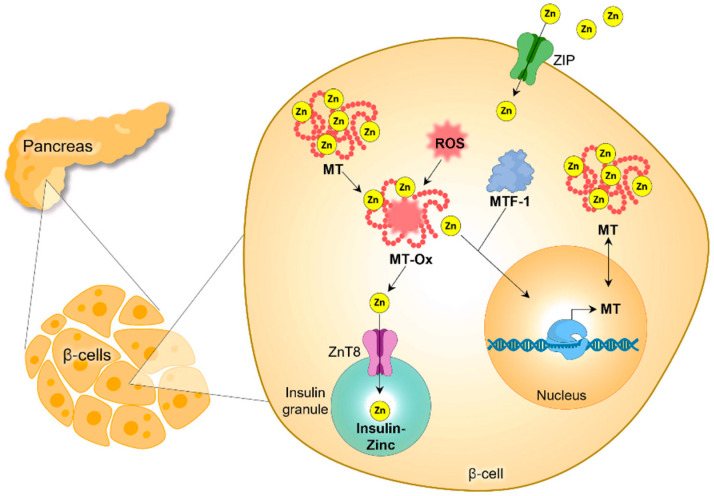
Zinc homeostasis and pancreatic β-cell: intracellular free cytosolic zinc can be imported via ZIP transporters or released from metallothionein (MT) during cellular stress. It enters insulin granules through ZnT8 to form crystalline insulin-zinc hexamers. Zinc also binds to metal-responsive transcription factor-1 (MTF-1), which translocates to the nucleus to up-regulate gene expression of MT. MT-Ox, oxidized MT; ROS, reactive oxygen species [62,63].

**Figure 5 cells-13-01359-f005:**
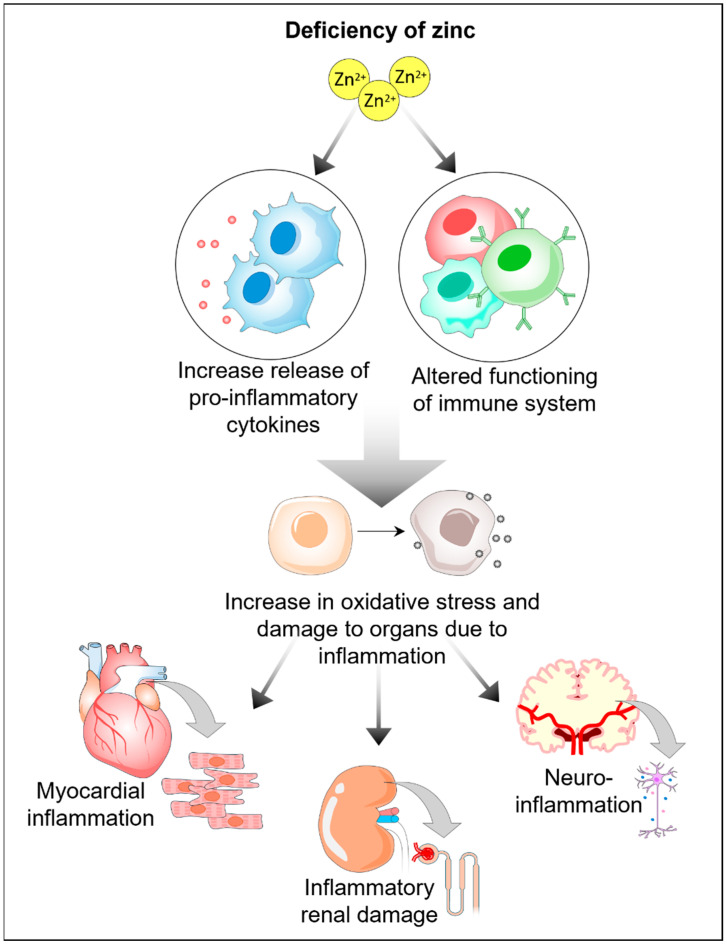
Zinc deficiency can induce an inflammatory microenvironment to induce systemic organ damage.

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
