# Peer review of "Zinc and Diabetes: A Connection between Micronutrient and Metabolism"

_cells, 2024, doi:10.3390/cells13161359_

Round 1

Reviewer 1 Report

Comments and Suggestions for Authors

Ahmad et al. summarized the role of zinc in glucose homeostasis and diabetes mellitus from zinc homeostasis to the regulation of zinc in insulin action and glucose regulation. The authors present clearly, and it is easy to follow. I have several comments as following:

1.      The authors mainly focus on the regulation of zinc in type 2 diabetes. I suggest that the authors clarify this point in the section “Objective of the study and methodology”.

2.      In section “The importance of maintaining optimal zinc balance in human health”, I feel that this section title is too broad, which distracts my attention from the focus of this manuscript. In this section, the authors mainly discuss how zinc regulation inflammation. Chronic inflammation is highly associated with diabetes mellitus, especially type 2 diabetes. This section provides an important molecular mechanism by which zinc regulates diabetes mellitus. The authors can emphasize diabetes mellitus instead of human health in the title.

3.      In line 168-189, the authors mentioned several inflammatory pathways regulated by zinc. The authors should include in which cell or organ and under what kind of conditions these pathways are detected.  In vitro or in vivo?

4.      Obesity is highly associated with type 2 diabetes. The authors should discuss the role of zinc on lipid metabolism during obesity.

5.      The SNPs on Zinc transporter-8 gene are associated with type 2 diabetes. The authors should discuss it in the manuscript.

6.      Sex and age are important factors for type 2 diabetes. Whether zinc homeostasis differs between sexes or at different ages?

Author Response

Ahmad et al. summarized the role of zinc in glucose homeostasis and diabetes mellitus from zinc homeostasis to the regulation of zinc in insulin action and glucose regulation. The authors present clearly, and it is easy to follow. I have several comments as following:

We sincerely thank the reviewer for their valuable feedback and for recognizing the significance of our manuscript. We greatly appreciate the time and expertise devoted to evaluating our work. In response to the reviewer's constructive comments, we have thoroughly revised the manuscript, carefully addressing each suggestion provided. The modified areas are highlighted in blue.

  1. The authors mainly focus on the regulation of zinc in type 2 diabetes. I suggest that the authors clarify this point in the section “Objective of the study and methodology”.

=> As correctly recommended by the reviewer, we’ve clarified this issue in the “Objective of the study and methodology” section.

  1. In section “The importance of maintaining optimal zinc balance in human health”, I feel that this section title is too broad, which distracts my attention from the focus of this manuscript. In this section, the authors mainly discuss how zinc regulation inflammation. Chronic inflammation is highly associated with diabetes mellitus, especially type 2 diabetes. This section provides an important molecular mechanism by which zinc regulates diabetes mellitus. The authors can emphasize diabetes mellitus instead of human health in the title.

=> Thank you for this suggestion, we’ve modified this section heading to reflect the content of the section.

  1. In line 168-189, the authors mentioned several inflammatory pathways regulated by zinc. The authors should include in which cell or organ and under what kind of conditions these pathways are detected.  In vitro or in vivo?

=> As the reviewer noted, we have included more information to clarify the conditions under which the experiments were conducted.

  1. Obesity is highly associated with type 2 diabetes. The authors should discuss the role of zinc on lipid metabolism during obesity.

=> We appreciate this suggestion, as recommended, we’ve briefly highlighted this issue in our revised manuscript.

  1. The SNPs on Zinc transporter-8 gene are associated with type 2 diabetes. The authors should discuss it in the manuscript.

=> We want to thank the reviewer for this suggestion, we’ve briefly discussed such association in our revised draft.

  1. Sex and age are important factors for type 2 diabetes. Whether zinc homeostasis differs between sexes or at different ages?

=> As correctly pointed out by the reviewer, we’ve briefly discussed this issue in our revised manuscript.

Reviewer 2 Report

Comments and Suggestions for Authors

This review focused on the role of zinc in glucose metabolism and the effects of its inadequacy on the development, progression, and complications of diabetes mellitus. Please conduct the concerns below.

1.      Zinc is an essential trace element and dose-dependent effect is varied. Therefore, it needs to conduct this in each discussion.

2.      Many metalloenzymes require zinc for their effects. The essential dose of zinc is important.

3.      The various zinc compounds used for supplementation included zinc chloride, zinc gluconate, zinc acetate, zinc phosphate, and zinc sulfate. Please describe the difference between them in application.

4.      Zinc supplements in commercial were not conducted. Why? Zinc supplementation at an optimal dose may provide beneficial effects on diabetic disorders. What is the optimal dose?

5.      The zinc supplementation on glycemic control needs more clinical data to support.

6.      Limitation(s) of current report may strengthen it.

Author Response

This review focused on the role of zinc in glucose metabolism and the effects of its inadequacy on the development, progression, and complications of diabetes mellitus. Please conduct the concerns below.

We extend our sincere gratitude to the reviewer for recognizing the significance of our manuscript. In response to their valuable feedback, we have thoroughly revised the document, carefully addressing each suggestion provided. To facilitate review, all modifications have been highlighted in blue throughout the manuscript.

  1. Zinc is an essential trace element and dose-dependent effect is varied. Therefore, it needs to conduct this in each discussion.

=> As correctly pointed out by the reviewer, we’ve briefly discussed the dose-dependent effects of zinc in our revised draft.

  1. Many metalloenzymes require zinc for their effects. The essential dose of zinc is important.

=> As recommended, we’ve briefly discussed this issue in our revised manuscript.

  1. The various zinc compounds used for supplementation included zinc chloride, zinc gluconate, zinc acetate, zinc phosphate, and zinc sulfate. Please describe the difference between them in application.

=> We appreciate this important suggestion, and briefly discussed various forms of zinc and their utility in our revised draft.

  1. Zinc supplements in commercial were not conducted. Why? Zinc supplementation at an optimal dose may provide beneficial effects on diabetic disorders. What is the optimal dose?

=> As recommended, we’ve briefly discussed the optimal dose of zinc in diabetic disorders in our revised draft. Although, exact dose is not standardized yet, a meta-analysis recommended dose of 20 mg/day of zinc for diabetic patients. We’ve included the information in the revised draft.  

  1. The zinc supplementation on glycemic control needs more clinical data to support.

=> As correctly pointed out by the reviewer, we’ve added relevant information on the effects of zinc on glycemic control.

  1. Limitation(s) of current report may strengthen it.

=> We want to thank the reviewer for this suggestion, we’ve briefly pointed out the potential limitations of our draft.

Round 2

Reviewer 1 Report

Comments and Suggestions for Authors

The authors answered all my questions.